



**Deciphering the Chemical Forms of Gaseous Oxidized**
**Mercury in Florida, USA**
Jiaoyan Huang[1], Matthieu B. Miller[2], Eric Edgerton[3], Mae Sexauer Gustin [2]
[1] Institute for the Environment, University of North Carolina, Chapel Hill, 100 Europa
Drive, Suite 490, Chapel Hill, NC, 27517, United States
[2] Department of Natural Resources and Environmental Sciences, University of Nevada-
Reno,1664 N. Virginia Street, Reno, NV, 89557, United States
[3] Atmospheric Research & Analysis, Inc., 410 Midenhall Way, Cary, North Carolina
27513, United States



**Abstract**
The highest mercury (Hg) wet deposition in the United States of America (USA) occurs
along the Gulf of Mexico, and in the southern and central Mississippi River Valley.
Gaseous oxidized Hg (GOM) is thought to be a major contributor due to high water
solubility and reactivity. Therefore, it is critical to understand concentrations, potential
for wet and dry deposition, and GOM compounds present in the air. Concentrations and
dry deposition fluxes of GOM were measured and calculated for Outlying Landing Field
(OLF), Florida, using data collected by a Tekran® 2537/1130/1135, the University of
Nevada-Reno Reactive Mercury Active System (UNRRMAS) with cation exchange and
nylon membranes, and Aerohead samplers that use cation-exchange membranes to
determine dry deposition. Relationships with Tekran® derived data must be interpreted
with caution, since GOM concentrations measured are biased low depending on the
chemical compounds in air, and interferences with water vapor and ozone.  Criteria air
pollutants were concurrently measured.
Results from nylon membranes with thermal desorption analyses indicated five GOM
compounds in this area, including $HgBr_2$, HgO, Hg-nitrogen and sulfur compounds, and
an unknown compound. This indicates that the site is influenced by different gaseous
phase reactions and sources. Using back trajectory analysis a high GOM event related to
high CO, but average $SO_2$, indicated air parcels moved from the free troposphere, and
across Arkansas, Mississippi, and Alabama at low elevation (<300 m). This event was
initially characterized by $HgBr_2$ followed by a mixture of GOM compounds. GOM
chemistry indicates reactions with local mobile source pollutants and long range transport
from outside of the USA.
In order to develop methods to measure GOM concentrations and chemistry, and model
dry deposition processes, the actual GOM compounds need to be known, as well as their
corresponding physicochemical properties, such as Henry's Law constants.
*Keywords: multiple-resistance model, dry deposition, cation exchange membrane,*
*criteria pollutants, active samplers*



## 1 Introduction


Mercury (Hg) has been classified as a persistent, bioaccumulative toxin (PBT) (UNEP,
2013), and deposition from the atmosphere is considered the dominant pathway by which
Hg enters remote ecosystems (Lindberg et al., 2007). In some areas, scavenging by
precipitation controls atmospheric Hg removal processes, such as in the Southeastern
United States of America (USA), where precipitation amounts are high (Prestbo and Gay,
2009). However, wet deposition concentrations are not necessarily correlated with
precipitation amounts >81mm, and deposition did not decreased with emission reductions
as coal combustion facilities in the region have implemented control technologies
(Prestbo and Gay, 2009). A contributing factor to wet deposition in the Gulf Coast area
may be related to high atmospheric convection during thunderstorms and scavenging of
gaseous oxidized Hg (GOM) from the free troposphere (Nair et al., 2013), and down
mixing of air with high GOM from the free troposphere (cf. Gustin et al., 2012).
An additional concern is that the Tekran® system measurement currently used to
quantify GOM does not equally quantify all GOM forms, and has interferences with
water vapor and ozone (cf. Ambrose et al., 2013; Gustin et al., 2013; Huang et al., 2013;
Lyman et al., 2010; McClure et al., 2014; Lyman et al., 2016 ). Since GOM is considered
an important form that can be rapidly removed from the atmosphere due to high water
solubility (Lindberg et al., 2007); it is important to understand both atmospheric
concentrations and chemistry (i.e. specific chemical compounds). Use of the University
of Nevada-Reno Reactive Mercury Active System (UNRRMAS) that collects GOM on
nylon membranes in tandem with cation exchange membranes has indicated that there are
different chemical compounds in the air and concentrations are 2 to 13 time higher than
previously thought (Huang et al., 2013; Gustin et al., 2016).
Mercury has been studied in Florida for many years, initially because of the high
concentrations measured in fish and the Florida Panther (Dvonch et al., 1999; Gustin et
al., 2012; Marsik et al., 2007; Pancras et al., 2011; Peterson et al., 2012). Long-term
GEM and GOM concentrations as measured by the Tekran® system have declined;
however, PBM concentrations increased after 2009 (Edgerton, unpublished data),
suggesting the atmospheric chemistry has changed. Peterson et al. (2011) and Gustin et al.





(2012) suggested based on detailed assessment of passive sampler and Tekran® system
collected Hg data, criteria air pollutants, and meteorology that at 3 locations in Florida
different GOM compounds were present and these were generated by *in situ* oxidation
associated with pollutants generated by mobile sources, indirect and direct inputs of Hg
from local electricity generating plants, and direct input of Hg associated with long range
transport. At OLF, background deposition was equal to that associated with mobile
sources, and a significant component was derived from long range transport in the spring.
In this work, GOM collected using the UNRRMAS, and the Aerohead dry deposition
measurement method (Lyman et al., 2007; 2009) were analyzed, along with Tekran® Hg
and criteria air pollutant data to understand GOM chemistry and dry deposition at
Outlying Landing Field (OLF), located ~ 15 kilometers NW of Pensacola, Florida.
Mercury was measured concurrently with various trace gases (CO, $O_3$, $SO_2$, NOx, NOy)
and meteorology. Air GEM/GOM/PBM concentrations were measured by the Tekran®
2537/1130/1135 system, respectively.
GOM dry deposition fluxes were calculated using deposition velocities determined using
a multi-resistance model with ambient air GOM concentrations from the Tekran®
system (multiplied by a factor of 3 due to bias in the Tekran® system; cf. Huang and
Gustin, 2015), and compared to those obtained using Aerohead data. Results were used to
estimate dry deposition velocities for the GOM compounds observed. The hypothesis for
this work was that since GOM compounds can vary spatially and temporally, due to
different compounds produced by different sources and processes, this will result in
different dry deposition velocities and dry deposition flux.
**2 Methods**
2.1 Field site
The sampling site was located at OLF (30.550°N, 87.374°W, 44 m above sea level). The
closest major Hg emission source is a coal-fired power plant (Plant Crist) northeast of the
site (Figure 1). This area has been used for atmospheric Hg research in previous studies
(Caffrey et al., 2010; Lyman et al., 2009; Gustin et al., 2012; Peterson et al., 2012; Weiss-





Penzias et al., 2011). OLF is a coastal site (~25 km away from Gulf of Mexico)
influenced by sea breezes especially during the summer (Gustin et al., 2012). Based on
cluster analyses of data from one year at this location ~24% of the air comes from the
marine boundary layer during the day and 60% during the night (Figure 1).
2.2 Sampling Methods
Aerohead samplers for determination of dry deposition were deployed bi-weekly from
June 2012 to March 2014. UNRRAMS were deployed bi-weekly from March 2013 to
March 2014. Atmospheric Hg concentrations, including GEM, GOM, and PBM, were
measured using a Tekran® system (model 2357/1130/1135, Tekran® Instrument Corp.,
Ontario, Canada) that was operated with one-hour sampling and one-hour desorption with
detection limits of 0.1 ng m$^{-3}$, 1.5 pg m$^{-3}$, and 1.5 pg m$^{-3}$, respectively.
Reactive Hg (GOM + PBM) concentrations were measured using the UNRRAMS with 3
sets of two in-series 47 mm cation-exchange membranes (ICE450, Pall Corp., MI, USA).
Three sets of nylon membranes (0.2 μm, Cole-Parmer, IL, USA) were also deployed to
assess Hg compounds in the air.  Cation exchange membranes have been demonstrated to
quantitatively measure specific compounds of GOM in the laboratory; however, may not
measure all compounds (Gustin et al., 2015; Gustin et al., 2016). Nylon membranes do
not retain GOM compounds quantitatively, and retention during transport needs to be
tested (Huang et al., 2013; Gustin et al., 2015; Gustin et al., 2016). Nylon membrane
retention is impacted by relative humidity that might limit uptake of specific forms.
Criteria air pollutants and meteorological data, including CO, SO$_2$, O$_3$, PM$_{2.5}$, NO, NO$_2$,
NO$_y$, temperature, relative humidity, wind speed, wind direction, pressure, solar radiation,
and precipitation were available at this site for the sampling period.  See Peterson et al.
( 2012) for detailed information on collection of these measurements.
Aeroheads and membranes were prepared at UNR, packed in a thermal isolated cooler,
and shipped back and forth between the laboratory and site. Samples were stored in a
freezer (-22°C) at UNR until analyzed. Cation-exchange membranes were digested and
analyzed following  EPA Method 1631 E (Peterson et al., 2012), and nylon membranes
were first thermally desorbed, and then analyzed using EPA Method 1631 E (Huang et al.,





2013). Cation-exchange membrane blanks for Aerohead and UNRRAMS were 0.40±0.18
(n=42), 0.37±0.26 (n=77) ng, respectively; and for nylon membranes used in the active
system blanks were 0.03±0.03 (n=69) ng. Therefore, method detection limits (MDL, 3-
sigma) for two-week sampling time (336 hrs) was 0.13 ng m$^{-2}$ hr$^{-1}$ for dry deposition,
respectively. For the active membrane system, the Hg amount on the back-up filters and
blanks were not significantly different (cation-exchange membrane: 0.4±0.3 vs 0.4±0.3
ng; nylon membrane: 0.03±0.03 vs 0.02±0.02 ng); therefore, the back-up filters were
included in the calculation of the bi-weekly blanks. The bi-weekly MDL (336 hrs) for
active system with cation-exchange and nylon membranes were 2-68 pg m$^{-3}$(mean: 24 pg
m$^{-3}$) and 0.01-14.6 pg m$^{-3}$(mean: 2.1 pg m$^{-3}$), respectively. Bi-weekly MDL was
calculated from 3 times the standard deviation of bi-weekly blanks. The MDL was
calculated for each period of sampling, due to the fact this can vary based on treatment of
the membranes, when preparing samples for deployment, deployment at the field site,
and handling once returned to the laboratory. The membranes may also vary by material
lot. All samples were corrected by subtracting the blank for the corresponding two-week
period.
2.3 Data analyses
Hourly Tekran®, criteria air pollutants, and meteorological data were managed and
validated by Atmospheric Research & Analysis, Inc (see Peterson et al., 2012). These
were then averaged into two-week intervals to merge with the membrane measurements.
In previous studies, scaling factors similar to HNO$_3$ ($\alpha=\beta=10$) were used to calculate
oxidized Hg dry deposition velocity (Marsik et al., 2007; Castro et al., 2012); however,
Lyman et al. (2007) used the effective Henry's Law constant, and half-redox reactions in
neutral solutions of HgCl$_2$, and indicated HONO might better represent the chemical
properties of oxidized Hg rather than HNO$_3$. Huang et al. (2015a) indicated that due to
limited understanding of oxidized Hg chemical properties, no single value can be used to
calculate oxidized Hg dry deposition, because $\alpha$ and $\beta$ would change with different GOM
compounds. Here dry deposition was calculated using the multiple resistance model of
Lyman et al. (2007) using both $\alpha=\beta=2, 5, 7,$ and 10.



Back trajectories were calculated using the Hybrid Single Particle Lagrangian Integrated
Trajectory (HYSPLIT 4.9) with EDAS 40-km, 1000-meter starting height. For day and
nighttime analyses, starting times were Local Standard Time (LST) 1100-1300 h and
100-300 h, 24-hour simulations. For a high-concentration event analyses, trajectories
were started for each day at 0000, 0400, 0800, 1200, 1600, and 2000 LST. Overall, the
uncertainties of back trajectories calculated from HYSPLIT are 20% of the air parcel
traveling distance (Draxler, 2013 ; Gebhart et al., 2005; Stohl, 1998; Stohl et al., 2003).
Back trajectories for the entire sampling time were analyzed using cluster analysis (Liu et
al., 2010).
Sigmaplot 14.0 (Systat Software Inc, San Jose, CA, USA), and Minitab 16.0 (Minitab
Inc., PA, US) were used to do t-tests and correlation analyses. Comparisons were
considered significantly different and correlations considered significant when $p < 0.05$.
**3 Results and Discussion**
3.1 Overall measurements
Similar to previous work at this location (Gustin et al., 2012), $O_3$ was highest in the
spring, and CO concentrations were high in winter due to a low boundary layer and
biomass burning, and low in summer (Table 1). Observations from the 3 GOM sampling
methods (Tekran®, and nylon, and cation exchange membranes) showed higher GOM
concentrations in spring relative to other seasons (Table 1). Concentrations of GOM
measured by cation-exchange membranes in the active system were significantly (p-value
< 0.05, paired-t test) higher than those measured by Tekran® KCl-coated denuder and
nylon membranes, both of which have been reported to be influenced by relative
humidity (Huang and Gustin, 2015b; Gustin et al., 2015). Mean cation-exchange
membrane concentrations were higher than Tekran® derived GOM by 14, 48, 11, and 13
times in the spring, summer, fall and winter, respectively.
Nylon membranes collected higher GOM concentrations than those measured by the
Tekran® in spring 2013 when the humidity was low. Overall, air concentrations
measured by the Tekran® system in this study were similar to those measured in 2010



(Peterson et al., 2012). Particulate-bound Hg had the same diel trend as GOM, but higher
concentrations.
Understanding the oxidants present in air is important for understanding potential GOM
compounds. Oxidants to consider include $O_3$, halogenated compounds, sulfur and
nitrogen compounds (cf. Gustin et al., 2016). Since the active system is currently limited
to 2-week sampling period it is difficult to use the data collected to determine specific
sources; however, they are useful for understanding the specific compounds that might be
present, and this in turn can be used to understand sources.
3.2 Potential GOM compounds
Standard desorption profiles for GOM compounds obtained by Huang et al. (2013) and
Gustin et al. (2015) are compared to those obtained at OLF (Figure 2). Compounds in the
permeation tubes included $HgBr_2$, $HgCl_2$, $HgN_2O_6 \cdot H_2O$, $HgSO_4$, and HgO. $HgCl_2$ and
$HgBr_2$ have been identified as being released from permeation tubes (Lyman et al., 2016);
however, the exact N and S compounds are not known. Only during 10 periods the nylon
membranes (collected in triplicate) collected a significant amount of GOM based on their
bi-weekly detection limit (Figure 2), and their desorption profiles varied. Although data
are limited, because we have observed similar thermal desorption compounds in other
studies (i.e. Huang et al, 2013 and Gustin et al. 2016), this indicates different chemical
forms are being collected, and that the compounds are not being generated on the
membranes. This has been shown to be the case in a limited study (Pierce and Gustin,
2016). In addition to our work, two forms of GOM in Montreal Canada air were reported
(Deeds et al., 2015).
Five distinct patterns of release were observed during thermal desorption. One had a
high residual tail that does not match our standard profiles; however, was also observed
in Nevada (Gustin et al, 2016). These occurred on 4/2/2013, 4/9/2013 and 5/21/2013.
This suggests that in spring there is a compound that is unknown based on current
standard profiles. A nitrogen-based compound was found on 5/21/2013 based on the
desorption profile. The second pattern occurred on 3/19/2013 and 11/19/2013, and this





corresponds to $HgBr_2$ with some residual tail that is again some compound not accounted
for.

The third pattern that occurred on 5/7/2013 and 8/27/2013, and corresponds to Hg-
nitrogen based compound with a residual tail. The $4^{th}$ pattern occurred on 1/14/2014, and
9/24/2013 was associated with $HgSO_4$ and the error bars are small. Lastly, the data
collected on 10/22/23 was noisy and had subtle peaks that correspond with HgO, a
nitrogen-based compound, and a high residual tail. It is interesting to note that the 11/19
profile was similar to $HgCl_2$.
Previous studies reported consistent desorption profiles from 3 sites in Nevada and
California without significant point sources (Huang et al., 2013). Huang et al. ( 2013)
presented desorption profiles from a highway, agriculture, and marine boundary layer site.
Profiles from the marine boundary layer and agriculture impacted site did not show clear
residual tails at 185ºC, but these were observed at the highway impacted site. In addition,
at OLF, a significant amount of GOM (15-30%) was released after 160 ºC.  This implies
that we are missing one or more GOM compound(s) (Figure 2) in our permeation profiles.
Interestingly, a peak was found in the 4/9/2013 sample at the GEM release temperature,
and this is not due to GEM absorption as demonstrated by Huang et al.(2013), and was
also observed in Nevada (Gustin et al., 2016), suggesting an additional unidentified
compound. This information indicates GOM compounds at OLF varied with time, and
this variation is due to complicated Hg emission sources and chemistry at this location (cf.
Gustin et al., 2012).
At OLF, GOM composition on the nylon membrane was  more complicated than that
collected at rural sites in the Western US (cf. Huang et al.,2013; Gustin et al., 2016);
however, similar complexity was observed at a highway location in Reno, Nevada
(Gustin et al., 2016). Desorption curves from the nylon filters collected at rural locations
in Nevada were in the range of the standard GOM compounds that have been investigated
(Huang et al., 2013: Gustin et al., 2016). Curves with multiple peaks in this study imply
that there were at least 5 GOM compounds collected on the nylon membranes.
3.3 Dry deposition measurements
Dry deposition of GOM measured by the Aerohead sampler ranged from 0 to 0.5 ng m$^{-2}$
hr$^{-1}$, and 83% of GOM dry deposition was higher than the detection limit (0.12 ng m$^{-2}$ hr$^{-}$
$^{1}$). Higher GOM dry deposition was observed in spring relative to winter (ANOVA one-
way rank, p-value < 0.01); GOM dry deposition was slightly lower in summer and fall
(not statistically different) relative to the  spring due to high wet deposition and
scavenging processes during these seasons. The pattern in GOM seasonal dry deposition
was similar to that reported by Peterson et al. (2012). However, GOM dry deposition
rates were significantly higher in this study than 2010 values (0.2 vs 0.05 ng m$^{-2}$ h$^{-1}$).
This is due to the correction of 0.2 ng m$^{-2}$ h$^{-1}$ applied in Peterson et al. (2012) to account
for contamination of the Aerohead that has been demonstrated to be unnecessary (Huang
et al., 2014). Although, highest GOM dry deposition measured using the Aerohead
sampler and GOM concentrations measured using the UNRRAMS were observed in
spring 2013, the value in March 2014 was relatively low. In March 2014, atmospheric
conditions were more similar to winter than spring, with low temperatures and high CO
concentrations. These results are different from those calculated using Tekran®
measurements that suggest low GOM concentrations and high deposition velocities, and
this is because the denuder measurements are biased low.
Modeled GOM dry deposition fluxes were calculated using GOM concentrations
measured by the Tekran® system that were multiplied by a factor of three (cf. Huang et
al., 2014). In general, measured Hg dry deposition fluxes were similar to those modeled
simulations with modeled GOM dry deposition α=β=2 during winter, spring, and fall (see
below; Figure 3). However, measured Hg dry deposition was significantly higher than
modeled results (both α=β=2 and 10) in summer and early fall (Figure 3). This indicates
there are compounds of GOM in the summer that are poorly collected by the denuder,
and this also can help explain the higher wet deposition measured during this season
(Prestbo and Gay, 2009). Highest deposition was measured during the spring, when the
input from long range transport is greatest (Gustin et al., 2012). Figure 3 shows the
disparity that occurs by season, and comparing model and measured vales. For example



in spring a=b=10 significantly overestimates deposition, while in the summer and early
fall measured deposition is greater than modeled values.
Because of the low GOM concentrations and influence of humidity on the nylon
membrane measurements (Huang and Gustin, 2015b), GOM compounds were identified
only in one summertime sample as $HgN_2O_6 \cdot H_2O$. During this time, measured GOM dry
deposition was ~6 times higher than both modeled results, and considering the Tekran®
correction factor of 3, membrane-based $HgN_2O_6 \cdot H_2O$ dry deposition flux was ~18 times
higher than the Tekran®-model-based value. Gustin et al. (2015) indicated $HgN_2O_6 \cdot H_2O$
collection efficiency on cation-exchange membrane in charcoal scrubbed air was ~ 12.6
times higher than on Tekran (KCl-coated denuder.
However, in May 2013, two samples were dominated by a profile similar the Hg
nitrogen-based compound with lower measured/modeled ratios (2.1-6.0 with Tekran®
correction factor). This might be due to ambient air GOM chemistry being dominated by
a compound with a different dry deposition velocity, less interference on the denuder
surface, or parameters in the dry deposition scheme. In May, GOM concentrations
measured by the Tekran® were higher than in summer due to lower wet deposition and
mean humidity (Table 1). Therefore, despite the fact that GOM collection efficiency
associated with the Tekran and nylon membranes are impacted by environmental
conditions, this demonstrates the presence of different compounds in the air. The dry
deposition scheme needs Henry's Law constants for determining the scaling factors for
specific resistances for different compounds (Lyman et al., 2007; Zhang et al., 2002).
Lin et al. (2006) stated that the dry deposition velocity of  HgO is two times higher than
that for $HgCl_2$, due to the different Henry's Law constant. The Henry's Law constants for
$HgCl_2$, $HgBr_2$, and HgO presented in previous literature (Schroeder and Munthe, 1998)
have high uncertainty, for how these calculations were done is not clear (S. Lyman, Utah
State University, personal communication, 2015), and the constants for $HgN_2O_6 \cdot H_2O$ and
$HgSO_4$ are unknown. Some researchers considered that GOM is similar to $HNO_3$
($\alpha = \beta = 10$), and some treated GOM as HONO ($\alpha = \beta = 2$) (Castro et al., 2012; Lyman et al.,
2007; Marsik et al., 2007); however, using the parameters of $HNO_3$ could overestimate



GOM dry deposition velocities due to the differences of effective Henry's law constants
($HgCl_2$: ~$10^6$ $HNO_3$: ~$10^{13}$ M $atm^{-1}$).
If the ratios ($HgBr_2$: 1.6, $HgCl_2$:2.4, $HgSO_4$: 2.3, HgO: 3.7, and $HgN_2O_6$•$H_2O$: 12.6) of
GOM concentrations measured by the Tekran® versus cation-exchange membranes for
different GOM permeated compounds (Gustin et al., 2015; Huang et al., 2013) are used
to correct Tekran® GOM data in this study, modeled GOM dry deposition (Figure 3) are
not correlated with measurements.  For example, on 3/9/2013 and 11/19/2013 (Figure 3),
GOM was dominated by $HgBr_2$ and $HgCl_2.$ Dry deposition of $HgBr_2$ from Aerohead
measurements and modeling were close to α=β=10; however, modeled and measured
$HgCl_2$ dry deposition were matched as α=β=2. Average deposition velocity for α=β=2
was 0.78 cm $s^{-1}$, and for α=β=10 is 1.59 cm $s^{-1}$, if we assume the model is right.  There
were three samples that were identified as Hg-nitrogen based compounds using nylon
membranes; however, the ratios of measurement and modeling $HgN_2O_6$•$H_2O$ dry
deposition were inconsistent over time. In spring, all modeled $HgN_2O_6$•$H_2O$ dry
deposition values were much higher than measured values; however, in summer,
measured and modeled $HgN_2O_6$•$H_2O$ dry deposition were similar as α=β=5 (Table 2).  If
you assume the dry deposition measurements made by the surrogate surfaces are accurate
then this demonstrates there are different forms that occur over time, and these will have
different deposition velocities as suggested by Peterson et al. (2012).
3.4 Elevated Pollution Event
In spring 2013, there was a time period when high concentrations of $O_3$, CO, and all Hg
measured (Figure 4). Figure 5 shows that during this time air masses traveled west to east
across the continent.  The air movement pattern is similar to that found in Gustin et al.
(2012) for OLF Class 2 events which had low $SO_2$ concentrations. During this 4-week
period, air parcels traveling to OLF were in the free troposphere and descended to the
surface (Figure 5). Although there are coal-fired power plants in the upwind area within a
500 km range (Figure 1), the low $SO_2$ concentrations, and elevated CO, $O_3$, and GOM
values were not from fossil fuel combustion. Gustin et al. (2012) also indicated that free
troposphere air impacted OLF. The first few endpoints for these trajectories indicate air





parcels entered North America at > 1000 m agl; therefore, there was transport of some air
measured during this time from the free troposphere. Ozone concentrations were also
similar to those measured in Nevada in the free troposphere at this time (Gustin et al.,
2014). It is important to note that the back trajectories are only for 72 hours and the ones
that subsided to surface levels in the Midwest were traveling fast. We hypothesize this is
a common event in the spring that represents free troposphere transport. The chemical
composition of this event suggests potential input from Asia as previously suggested for
numerous locations in Florida in the spring by Gustin et al. (2012). During this time
$HgBr_2$ was an important compound initially and then the profile was a gradual increase
with a high residual tail. This would suggest initial subsidence of air from the
stratosphere/troposphere (cf. Lyman et al., 2012) followed by a mixture of polluted air as
observed in the Western United States (c.f. VanCuren and Gustin, 2015)
**4 Conclusions**
The chemical forms of GOM in the atmosphere at OLF varied by season as suggested by
Gustin et al. (2012).  Five potential different GOM compounds were identified at OLF
using nylon membranes with thermal desorption analysis, including $HgBr_2$, $HgCl_2$, HgO,
Hg-nitrogen and sulfur compounds, and 2 unknown compounds. Given the long sampling
time detailed assessment of specific sources is difficult, but the presence of different
compounds indicate multiple sources and different GOM chemistry. Comparing modeled
and measured Hg dry deposition fluxes also demonstrate there are different forms in air
and this will affect dry deposition velocities. In order to improve our understanding of Hg
air-surface exchange, and measure GOM physiochemical properties of different GOM
compounds need to be understood.
**5 Acknowledgements**
The authors thank The Southern Company (project manager-John Jansen) for their
support, and Bud Beghtel for deploying and collecting our membranes and passive
samplers at OLF and managing this site in general. This work was also supported by
EPRI and a National Science Foundation Grant 1326074. We thank the following
students for coordinating shipment of membranes and passive samplers, analyses of the





membranes in the lab, and keeping the glassware clean (Keith Heidecorn, Douglas Yan,
Matt Peckham, Jennifer Arnold, Jen Schoener, and Addie Luippold).

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





Table 1 – Overall seasonal average of air species, GEM, PBM, GOM (measured using
three different methods) concentration, GOM dry deposition (DD), and meteorological
data at OLF.

| | 2012 | | | 2013 | | | | 2014 |
|---|---|---|---|---|---|---|---|---|
| | Summer | Fall | Winter | Spring | Summer | Fall | Winter | March |
| Ozone [ppb] | 30±15 | 30±12 | 29±11 | 38±12 | 24±12 | 26±11 | 27±10 | 35±12 |
| CO [ppb] | 143±38 | 161±35 | 167±41 | 165±36 | 139±35 | 156±33 | 167±35 | 183±33 |
| $SO_2$ [ppb] | 0.3±0.4 | 0.6±1.5 | 0.4±0.5 | 0.3±0.5 | 0.2±0.3 | 0.4±0.5 | 0.7±1.2 | 0.3±0.4 |
| NO [ppb] | 0.3±0.7 | 0.3±0.7 | 0.3±0.8 | 0.2±0.5 | 0.3±0.7 | 0.3±0.8 | 0.4±0.8 | 0.2±0.5 |
| $NO_2$ [ppb] | 2.4±2.4 | 3.0±2.7 | 3.0±3.1 | 2.0±2.3 | 2.2±2.1 | 3.1±2.9 | 3.2±3.0 | 2.3±2.8 |
| $NO_y$ [ppb] | 3.6±2.9 | 4.3±3.1 | 4.3±3.6 | 3.1±2.8 | 3.2±2.5 | 4.4±3.3 | 4.2±3.4 | 3.6±3.1 |
| GEM [ng $m^{-3}$][a] | 1.2±0.1 | 1.2±0.1 | 1.3±0.1 | 1.2±0.2 | 1.1±0.1 | 1.0±0.1 | 1.2±0.3 | 1.2±0.1 |
| GOM [pg $m^{-3}$][a] | 0.6±1.3 | 1.1±2.8 | 1.0±2.2 | 2.9±5.1 | 0.5±1.0 | 1.1±2.1 | 1.3±2.5 | 2.0±3.6 |
| PBM [pg $m^{-3}$][a] | 2.4±2.6 | 3.6±3.8 | 7.3±8.7 | 5.9±6.8 | 2.3±2.0 | 2.9±2.3 | 4.9±5.3 | 4.0±3.4 |
| GOM [pg $m^{-3}$][b] | - | - | - | 43±110 | 24±57 | 14±18 | 17±23 | 24±15 |
| GOM [pg $m^{-3}$][c] | - | - | - | 4±10 | 0.4±1.3 | 1.2±1.1 | 0.6±0.6 | 0.6±0.5 |
| GOM DD [ng $m^{-2}$ $hr^{-1}$] | 0.24±0.20 | 0.17±0.12 | 0.15±0.06 | 0.40±0.23 | 0.20±0.13 | 0.13±0.18 | 0.20±0.50 | 0.14±0.04 |
| WS [m s-1] | 2.1±1.2 | 2.1±1.0 | 2.8±1.7 | 2.9±1.8 | 2.0±1.1 | 2.1±1.1 | 2.5±1.3 | 2.5±1.5 |
| TEMP [°C] | 26±3 | 19±6 | 14±6 | 18±6 | 26±3 | 20±7 | 11±7 | 14±5 |
| RH [%] | 83±14 | 76±18 | 79±19 | 73±21 | 84±13 | 77±17 | 76±23 | 78±21 |
| SR [w $m^{-2}$] | 230±302 | 193±271 | 121±199 | 266±304 | 210±278 | 175±255 | 129±212 | 182±278 |
| Precipitation [mm] | 637 | 186 | 385 | 223 | 1010 | 254 | 357 | 183 |

[a]: Tekran data
[b]: cation-exchange membrane data
[c]: nylon membrane data



Table 2 – Modeled (multiple-resistance model) and measured (surrogate surfaces) GOM dry deposition (ng m$^{-2}$ hr$^{-1}$), GOM concentrations used to calculate for modeled results are from the Tekran® data and corrected by compounds' corresponding ratios from Gustin et al. (submitted). The sample with unknown compound is used the Tekran® data with correction factor of three (average ratio). The tentative GOM compounds are identified from nylon membrane results.

| Start date | Tentative GOM compound | Measured GOM dry deposition flux | Modeled GOM dry deposition $\alpha=\beta=2$ | Modeled GOM dry deposition $\alpha=\beta=5$ | Modeled GOM dry deposition $\alpha=\beta=7$ | Modeled GOM dry deposition $\alpha=\beta=10$ |
|---|---|---|---|---|---|---|
| 3/12/2013 | HgBr$_2$ | 0.50±0.06 | 0.34 | 0.49 | 0.54 | 0.58 |
| 3/26/2013 | unknown | 0.40±0.11 | 0.34 | 0.47 | 0.52 | 0.56 |
| 4/30/2013 | Hg(NO$_3$)$_2$ | 0.50±0.13 | 1.21 | 1.67 | 1.81 | 1.95 |
| 5/14/2013 | Hg(NO$_3$)$_2$ | 0.40±0.09 | 1.19 | 1.69 | 1.88 | 2.07 |
| 8/20/2013 | Hg(NO$_3$)$_2$ | 0.15±0.07 | 0.10 | 0.14 | 0.16 | 0.17 |
| 11/12/2013 | HgCl$_2$ | 0.08±0.03 | 0.11 | 0.16 | 0.17 | 0.19 |
| 1/7/2014 | HgSO$_4$ | 0.19±0.03 | 0.18 | 0.24 | 0.27 | 0.29 |



Figure Caption

Figure 1 – Sampling site and point sources (NEI 2011) map. Cluster trajectories for daytime (11:00-13:00) and nighttime (1:00-3:00).

Figure 2 – Desorption profiles from nylon membranes with standard materials in laboratory investigation (top) and field measurements. Whisker is 1 standard variation, and only present in the desorption peak. Note the Hg-nitrogen compound in the permeation tube was $HgN_2O_6 \cdot H_2O$.

Figure 3 – Measured and modeled GOM dry deposition fluxes, Tekran® data (correction factor of three) were used with multiple resistance models ($\alpha = \beta = 2$ and 10). Tentative GOM compounds were determined using the results from nylon membranes desorption.

Figure 4 – Temporal variation of GOM concentrations (mean ± standard deviation, bi-week average), outlined rectangle indicates a polluted event with high Hg, CO, and ozone concentrations. Data are missing for 3 weeks because it was not collected. Tekran data is presented when >75% of the data were available and membrane data are shown when above the method detection limit.

Figure5 – Results of gridded frequency distribution (left), light color indicates less endpoints in a grid. Altitude of 72-hr trajectories during the polluted event (3/12/2013-4/2/2013), light color of dots on left panel represents low altitude.





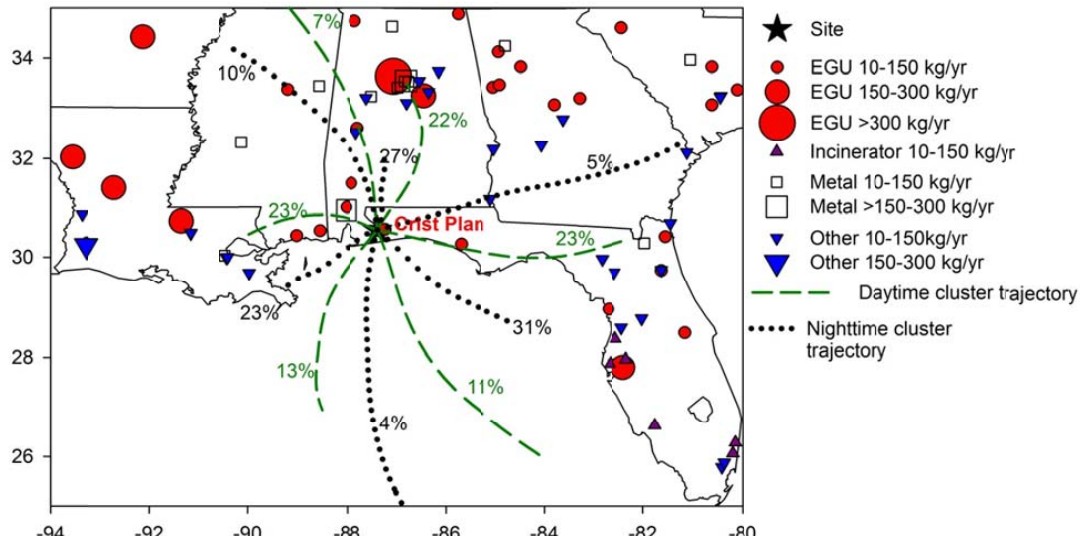

Figure 1 – Sampling site and point sources (NEI 2011) map. Cluster trajectories for daytime
(11:00-13:00) and nighttime (1:00-3:00).





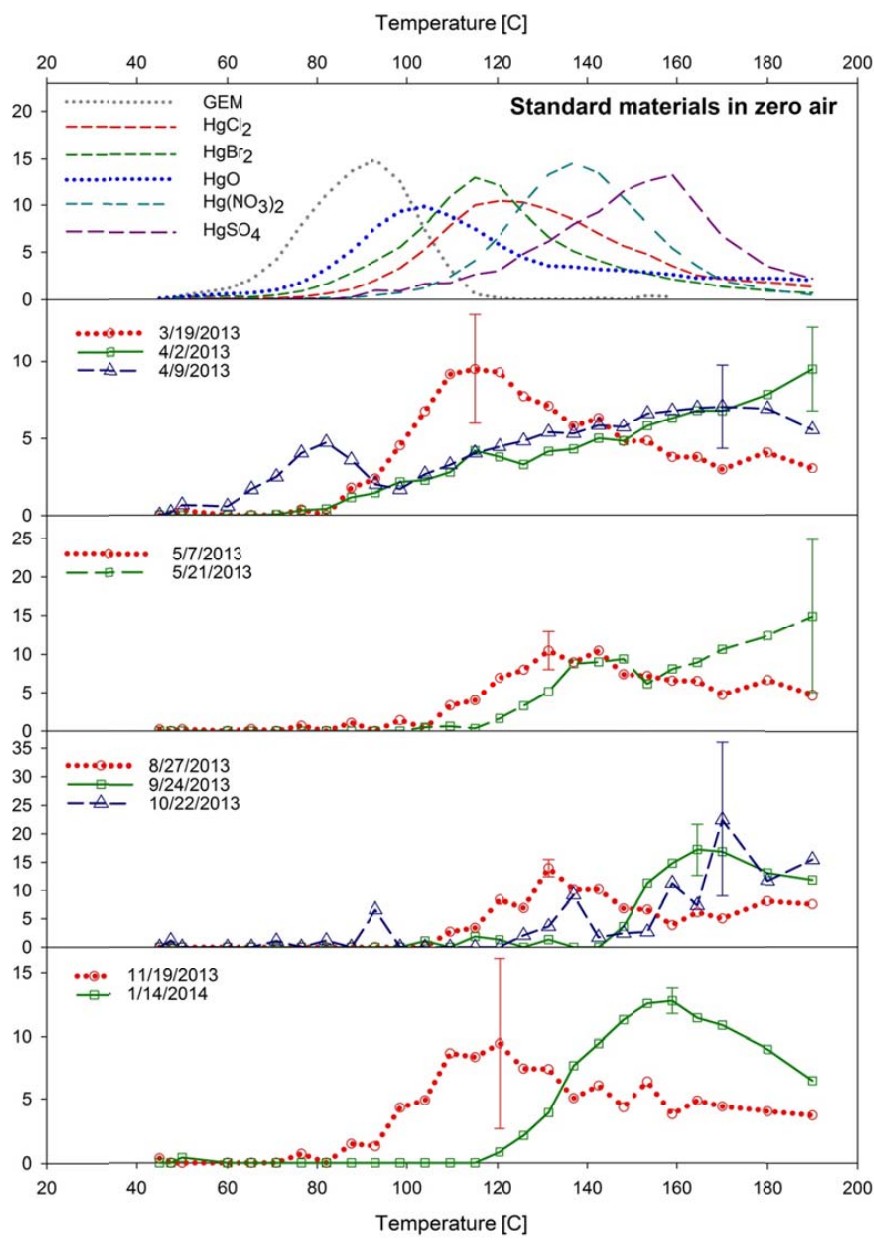

Figure 2 – Desorption profiles from nylon membranes with standard materials in laboratory investigation (top) and field measurements. Whisker is 1 standard variation, and only present in the desorption peak. Note the Hg-nitrogen compound in the permeation tube was $HgN_2O_6 \cdot H_2O$.



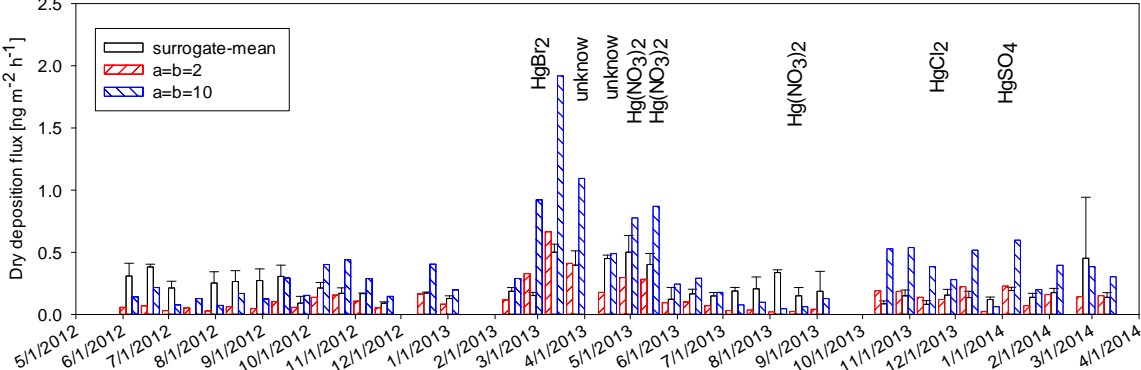

Figure 3 – Measured and modeled GOM dry deposition fluxes, Tekran® data (correction factor of three) were used with multiple resistance models (α=β=2 and 10). Tentative GOM compounds were determined using the results from nylon membranes desorption.



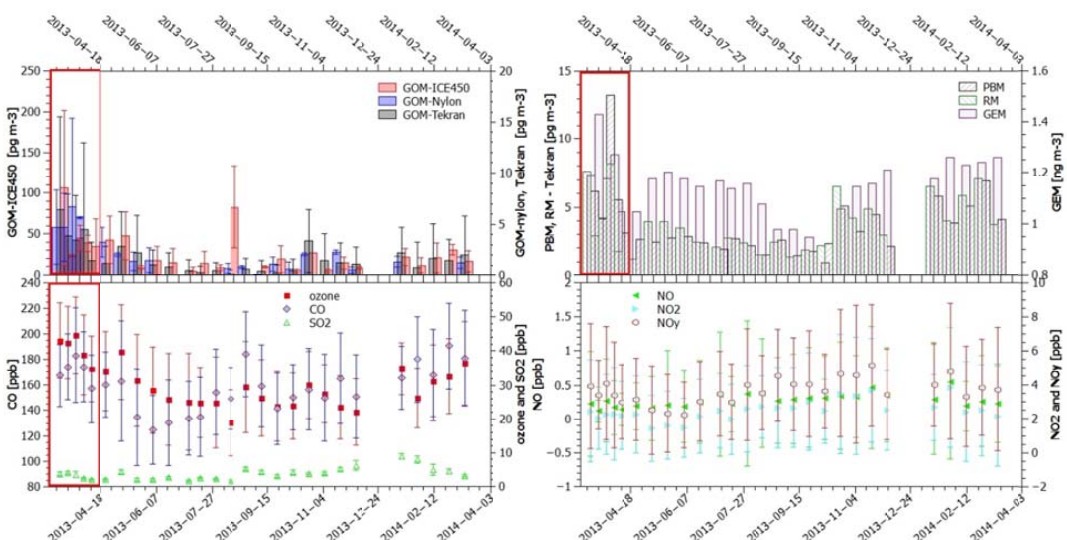

Figure 4 – Temporal variation of GOM concentrations (mean ± standard deviation, bi-week
average), outlined rectangle indicates a polluted event with high Hg, CO, and ozone
concentrations. Data are missing for 3 weeks because it was not collected. Tekran data is
presented when >75% of the data were available and membrane data are shown when above the
method detection limit.



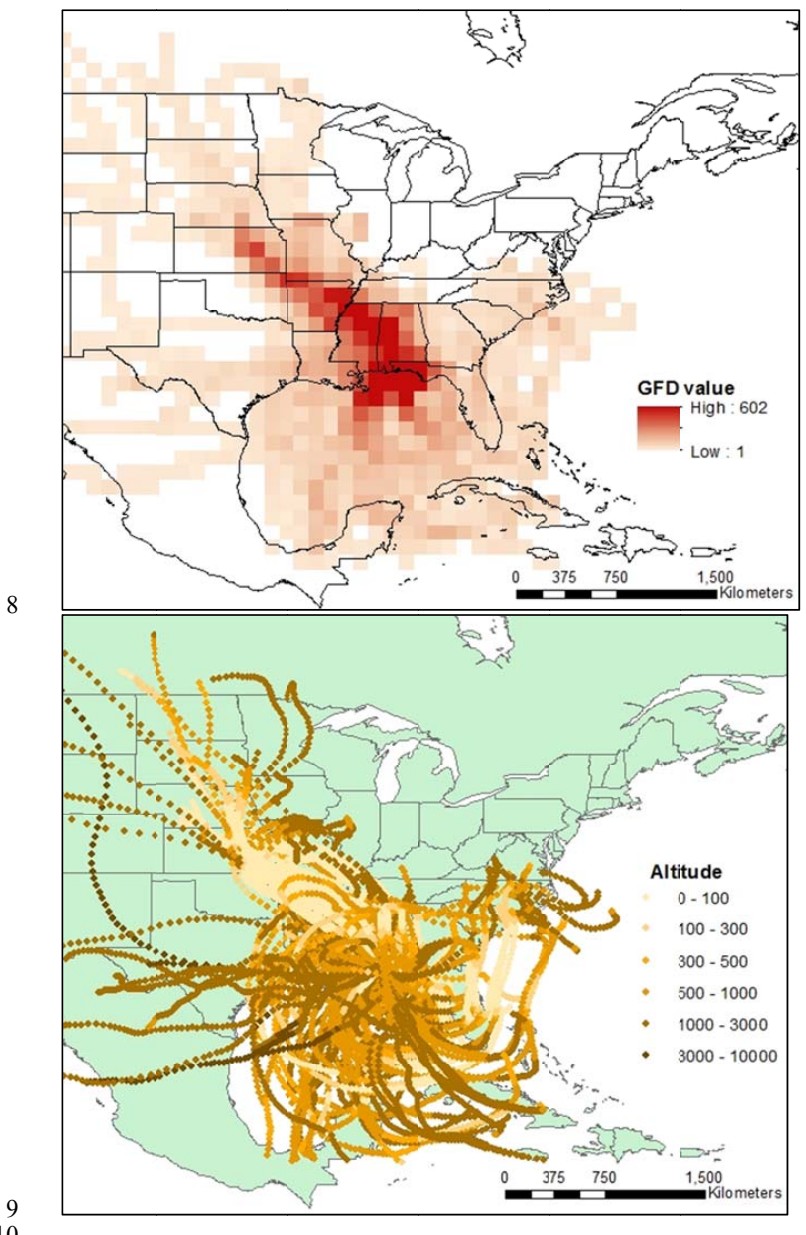

Figure 5 – Results of gridded frequency distribution (left), light color indicates less endpoints in
a grid. Altitude of 72-hr trajectories during the polluted event (3/12/2013-4/2/2013), light color
of dots on left panel represents low altitude.
