# Peer review of "Deciphering Potential Chemical Compounds of Gaseous"

_Atmospheric Chemistry and Physics, 2016_

## Referee Comment (RC1) · Anonymous Referee #2 · 8 Nov 2016

The authors present their latest findings regarding identification of potential different chemical forms of gaseous oxidized mercury (GOM) at a site in Florida. The information presented should be useful to those working in this field. Following are specific comments on the manuscript:

1. Recommend revising the title of the manuscript to: "Deciphering some potential chemical forms of gaseous oxidized mercury in Florida, USA"

2. For the unknown compound, recommend discussing in more detail potential candidates.

3. The abstract does not align with the Conclusions section in discussing the five potential different GOM compounds. For instance, the abstract does not mention HgCl2, and the Conclusions section mentions 2 unknown compounds while the abstract mentions

one unknown compound.

4. The Introduction, first paragraph, stated that deposition did not decrease with emission reductions as coal combustion facilities in the region (please clarify what region?) have implemented control technologies (Prestbo and Gay, 2009). The Prestbo and Gay is an older reference; would this still be the case in 2016?

5. In Section 3.2 Potential GOM Compounds, the end of the third paragraph ("it is interesting to note that the 11/19 profile was similar to HgCl2") does not align with the end of the second paragraph which describes HgBr2 instead of HgCl2.

---

## Author Comment (AC1) · 13 Nov 2016

Please note the Gustin et al. 2016 and Lyman et al. 2016 papers have been accepted and references updated

The authors present their latest findings regarding identification of potential different chemical forms of gaseous oxidized mercury (GOM) at a site in Florida. The information presented should be useful to those working in this field. Following are specific comments on the manuscript: 1. Recommend revising the title of the manuscript to: "Deciphering some potential chemical forms of gaseous oxidized mercury in Florida, USA" Response: Good comment we have changed this to "Deciphering potential chemical compounds of gaseous oxidized mercury in Florida, USA"

2. For the unknown compound, recommend discussing in more detail potential candidates.

Response: We have suggested the higher temperature ones could be organic compounds based on the methyl mercury desorption profile. Otherwise, we do not want to speculate. Line 212.

3. The abstract does not align with the Conclusions section in discussing the five potential different GOM compounds. For instance, the abstract does not mention HgCl2, and the Conclusions section mentions 2 unknown compounds while the abstract mentions one unknown compound.

Response: This has been adjusted to fit with the text. Thank you for catching that. There are 7potential compounds: 1) perhaps organic compounds when there is a high residual tail; 2) Hg-N and 3) Hg-S based compounds; 4) HgCl2 and 5) HgBr2; 6) HgO; and 7) something released at the same temperature as GEM.

4. The Introduction, first paragraph, stated that deposition did not decrease with emission reductions as coal combustion facilities in the region (please clarify what region?) have implemented control technologies (Prestbo and Gay, 2009). The Prestbo and Gay is an older reference; would this still be the case in 2016?

Response: Good question. Based on data presented by the Mercury Deposition Network, concentrations have gone up at OLF 17.1 in 2012 and 21.0 in 2014 http://nadp.sws.uiuc.edu/MDN/annualmdnmaps.aspx Site updated 2104 visited 13 November 2016.This information has been added to the introduction

5. In Section 3.2 Potential GOM Compounds, the end of the third paragraph ("it is interesting to note that the 11/19 profile was similar to HgCl2") does not align with the end of the second paragraph which describes HgBr2 instead of HgCl2. Response: This is now described as potentially being both.

---

## Referee Comment (RC2) · Anonymous Referee #1 · 5 Dec 2016

The manuscript presents field observations of ambient concentrations of speciated mercury and other air pollutants at a costal site in Florida. GOM were monitored with Tekran and membranes. GOM dry deposition fluxes were calculated using Tekran concentrations (with a correction factor) and dry deposition velocities, and monitored using surrogate surfaces. Some GOM forms were observed on some membranes, and those possible forms were used to estimate dry deposition fluxes with various correction factors. Other analyses include back trajectory. The topic is relevant to ACP. My comments/suggestions are listed below.

1) Major concern

My major concern is the originality and scientific contributions of this manuscript, con- sidering the large number of publications (as listed in the reference section of the

manuscript) by the two primary authors (i.e. Huang and Gustin) in the past few years on the topic of the chemical forms of GOM. The authors may want to clarify whether this manuscript presents results at additional sites, during a different time period, with different methodologies, or of different findings /concessions.

2) Editorial comments and suggestions

The use of English language is overall satisfactory. However, there is room for improvement.

a) L78-89, this section could be replaced by a brief summary of the methodology since a detailed description is presented in the Methods section.

b) There are a few awkward phrases and sentences, e.g. "∼24% of the air comes from the marine boundary layer during the day and 60% during the night"; "if we assume the model is right".

c) In the references, some papers are listed twice. Also, not sure if a manuscript under preparation (L440) could be included in ACP manuscripts.

d) Table 2, not sure if a manuscript under review could be included in ACP manuscripts.

e) Figure 5 caption was incorrect to me. The word "left" meant "top" then "bottom", suggest labelling subplots as a) and b).

―――――――――――――――

---

## Author Comment (AC2) · 7 Dec 2016

The manuscript presents field observations of ambient concentrations of speciated mercury and other air pollutants at a costal site in Florida. GOM were monitored with Tekran and membranes. GOM dry deposition fluxes were calculated using Tekran concentrations (with a correction factor) and dry deposition velocities, and monitored using surrogate surfaces. Some GOM forms were observed on some membranes, and those possible forms were used to estimate dry deposition fluxes with various correction factors. Other analyses include back trajectory. The topic is relevant to ACP.

Response: Thank you for reading our paper.

My comments/suggestions are listed below. 1) Major concern My major concern is the originality and scientific contributions of this manuscript, considering the large number of publications (as listed in the reference section of the manuscript) by the two primary authors (i.e. Huang and Gustin) in the past few years on the topic of the chemical forms of GOM. The authors may want to clarify whether this manuscript presents results at additional sites, during a different time period, with different methodologies, or of different findings /concessions.

Response: We have added two sentences to the abstract to clarify this. "Data were collected simultaneously using the UNRRMAS for the first time at this location and methods that have been applied before at OLF, allowing for comparison and better understanding of GOM. This work represents an attempt to measure the chemical compounds of GOM in the air."

2) Editorial comments and suggestions The use of English language is overall satisfactory. However, there is room for improvement. a) L78-89, this section could be replaced by a brief summary of the methodology since a detailed description is presented in the Methods section.

Response: Details noted in the methods were deleted.

b) There are a few awkward phrases and sentences, e.g. "_24% of the air comes from the marine boundary layer during the day and 60% during the night"; "if we assume the model is right".

Response: We have read through the paper and removed what we thought were awkward sentences.

c) In the references, some papers are listed twice. Also, not sure if a manuscript under preparation (L440) could be included in ACP manuscripts.

Response: This has been corrected and the manuscript under preparation removed.

d) Table 2, not sure if a manuscript under review could be included in ACP manuscripts.

Response: The Table caption has been updated with the correct references.

e) Figure 5 caption was incorrect to me. Response: The caption has been adjusted.
* * *

---

## Referee Comment (RC3) · Anonymous Referee #1 · 15 Dec 2016

The manuscript presents field observations of ambient concentrations of speciated mercury and other air pollutants at a costal site in Florida. GOM were monitored with Tekran and membranes. GOM dry deposition fluxes were calculated using Tekran concentrations (with a correction factor) and dry deposition velocities, and monitored using surrogate surfaces. Some GOM forms were observed on some membranes, and those possible forms were used to estimate dry deposition fluxes with various correction factors. Other analyses include back trajectory. The authors conclude that "the chemical forms of GOM in the atmosphere at OLF varied by season as suggested by Gustin et al. (2012)." However, it is unclear whether this manuscript presents results at additional sites, during a different time period, with different methodologies, or of different findings /conclusions from those reported in Gustin et al. (2012), Peterson et al. (2012), and the large number of publications (as listed in the reference section

of the manuscript) by the two primary authors (i.e. Huang and Gustin) in the past few years on the topic of the chemical forms of GOM. Furthermore, as a standalone publication, this manuscript should provide more data and theoretical support that "the chemical forms of GOM in the atmosphere at OLF varied by season." My specific comments/suggestions are listed below.

1. Major concerns

1.1 There is a lack of support to conclude that there are different forms of GOM in different seasons.

Due to the small number of samples and inconsistent identification of most GOM forms between samples, there is a large uncertainty in identification of most GOM forms. The same can be said with Tekran and passive samplers. Similarly, there are many factors contributing to the deviation between measured and model estimated fluxes with different adjustment factors. Furthermore, the notion of more knowledge of GOM forms would lead to a better model-measurement agreement is debateable, because it was built upon the assumption of only one GOM form present in each dry deposition or air sample. Therefore, the reviewer is not convinced that disagreement between the different methods is largely due to limitations of the Tekran system and different GOM forms at different times of the year, as the authors suggest, due to insufficient data supporting this.

In other words, the limitations of the sampling method and study design are inadequate to justify the arrival of the conclusions. By leaping from not observed to not present (in concentrations), then from poor model-measurement agreement (in fluxes) to the presence of existing but not observed forms, the chosen approach seems to be convenient rather than fully based on science. The authors may want to provide an in-depth discussion of any observational or theoretical considerations supporting seasonal variation of chemical forms of ambient GOM at OLF. For example, the changes in any of the following factors with season, 1) prevailing wind/air mass directions, 2) oceanic or

meteorological conditions, e.g. ambient temperature and ocean currents, 3) major local Hg sources, 4) major upwind Hg sources, 5) ambient concentrations of major reactants related with the GOM forms observed or hypothesized, and 6) any other factors that may lead to different GOM forms at OLF in different seasons. Furthermore, the authors may need to explain why any of those seasonal variations would lead to different chemical forms of ambient GOM at OLF. Perhaps the phrase "different predominant forms" is more conservative.

1.2 Section 3.4 Elevated Pollution Event: the selection of one elevated event raised many questions instead of establishing a valid association between the observed concentration levels and the observed GOM forms.

The analyses in this section rely on a few two-week samples and 72-hr back trajectories to "hypothesize this is a common event in the spring that represents free troposphere transport". The questions that need further explanation include: 1) why free troposphere transport is limited to spring if it is responsible for elevated concentrations, 2) if the prevailing air mass directions support the trans-pacific transport of air pollutants, why it only caused elevated concentrations in spring, 3) given the proximity of the sampling site to the ocean, why HgBr2 was only observed once in spring during the one year study period, and 4) a prolonged period of elevated pollutant concentrations is often associated with local emissions and/or local/regional weather conditions. What evidences suggest that this site was under "the impact of subsidence of air from the stratosphere/troposphere" for 4-weeks with little impact of synaptic weather conditions which could lead to strong mixing hence lower concentrations. The authors may want to include analyse of other periods when GOM forms were observed or omit this section considering the limitations of using two-week samples and 72-hr back trajectories to explain long range transport (beyond three days) and transformations of Hg (in the time scale of hours to days).

2. Editorial comments and suggestions

1) "Tekran$^{®}$, and nylon, and cation exchange membranes", please clarify.

2) L187-192, this paragraph is less relevant to section 3.1, could be omitted or placed in another section.

3) "Overall, air concentrations measured by the Tekran$^{®}$ system in this study were similar to those measured in 2010 (Peterson et al., 2012)." and L250, same site?

4) L200-204, please rephrase.

5) L205, please provide the two forms.

6) L236-238, please provide the forms at the rural site and traffic site.

7) "Modeled GOM dry deposition fluxes were calculated", "measured Hg dry deposition fluxes were similar to those modeled simulations with modeled GOM dry deposition $\alpha=\beta=2$", please rephrase.

8) L282-283, please rephrase.

9) "the profile was a gradual increase", please rephrase.

10) "In order to improve our understanding of Hg air-surface exchange, and measure GOM physiochemical properties of different GOM compounds need to be understood", please rephrase.

---

## Author Comment (AC3) · 18 Dec 2016

Please note the revised paper has been uploaded as a pdf.

 The manuscript presents field observations of ambient concentrations of speciated mercury and other air pollutants at a costal site in Florida. GOM were monitored with Tekran and membranes. GOM dry deposition fluxes were calculated using Tekran concentrations (with a correction factor) and dry deposition velocities, and monitored using surrogate surfaces. Some GOM forms were observed on some membranes, and those possible forms were used to estimate dry deposition fluxes with various correction factors. Other analyses include back trajectory. The authors conclude that "the chemical forms of GOM in the atmosphere at OLF varied by season as suggested by Gustin et al. (2012)." However, it is unclear whether this manuscript presents results at additional sites, during a different time period, with different methodologies, or of different findings /conclusions from those reported in Gustin et al. (2012), Peterson et al. (2012), and the large number of publications (as listed in the reference section of the manuscript) by the two primary authors (i.e. Huang and Gustin) in the past few years on the topic of the chemical forms of GOM.

Response: We have highlighted new information. Furthermore, as a standalone publication, this manuscript should provide more data and theoretical support that "the chemical forms of GOM in the atmosphere at OLF varied by season." My specific comments/suggestions are listed below.

Response: We thank this reviewer for her/his comments led us to clarify some important points.

1. Major concerns 1.1 There is a lack of support to conclude that there are different forms of GOM in different seasons.

Response: The dry deposition measurement data analyses here suggests this. See also Gustin et al., 2012 and Peterson et al. 2012.

Due to the small number of samples and inconsistent identification of most GOM forms between samples, there is a large uncertainty in identification of most GOM forms. The same can be said with Tekran and passive samplers. Similarly, there are many factors contributing to the deviation between measured and model estimated fluxes with different adjustment factors. Furthermore, the notion of more knowledge of GOM forms would lead to a better model-measurement agreement is debateable, because it was built upon the assumption of only one GOM form present in each dry deposition or air sample. Therefore, the reviewer is not convinced that disagreement between the different methods is largely due to limitations of the Tekran system and different GOM

forms at different times of the year, as the authors suggest, due to insufficient data supporting this.

Response: We have added discussion that improves our certainty regarding the thermal desorption measurements. Please see Huang et al., 2013 and other papers referenced.

In other words, the limitations of the sampling method and study design are inadequate to justify the arrival of the conclusions. By leaping from not observed to not present (in concentrations), then from poor model-measurement agreement (in fluxes) to the presence of existing but not observed forms, the chosen approach seems to be convenient rather than fully based on science. The authors may want to provide an in-depth discussion of any observational or theoretical considerations supporting seasonal variation of chemical forms of ambient GOM at OLF.

Response: Many of the factors below are discussed in Gustin et al., 2012. For example, the changes in any of the following factors with meteorological conditions, e.g. ambient temperature and ocean currents, 3) major local Hg sources, 4) major upwind Hg sources, 5) ambient concentrations of major reactants related with the GOM forms observed or hypothesized, and 6) any other factors that may lead to different GOM forms at OLF in different seasons. Furthermore, the authors may need to explain why any of those seasonal variations would lead to different chemical forms of ambient GOM at OLF. Perhaps the phrase "different predominant forms" is more conservative.

Response: In order to do this we would need more data which we do not have. As it is we begged for funding to collect this limited data set.

1.2 Section 3.4 Elevated Pollution Event: the selection of one elevated event raised many questions instead of establishing a valid association between the observed concentration levels and the observed GOM forms.

Response: We did not "select" anything. This was an event that happened in the spring.

The analyses in this section rely on a few two-week samples and 72-hr back trajectories to "hypothesize this is a common event in the spring that represents free troposphere transport". The questions that need further explanation include: 1) why free troposphere transport is limited to spring if it is responsible for elevated concentrations,

Response: Please see the recent literature on ozone (Gertler and Bennett, 2015; Lefohn and Cooper, 2015)

2) if the prevailing air mass directions support the trans-pacific transport of air pollutants, why it only caused elevated concentrations in spring, 3) given the proximity of the sampling site to the ocean, why HgBr2 was only observed once in spring during the one year study period, Response: It appears N and S compounds can be important in the marine boundary layer please see Huang et al (2013). We agree this is limited data.

and 4) a prolonged period of elevated pollutant concentrations is often associated with local emissions and/or local/regional weather conditions. What evidences suggest that this site was under "the impact of subsidence of air from the stratosphere/troposphere" for 4-weeks with little impact of synaptic weather conditions which could lead to strong mixing hence lower concentrations. The authors may want to include analyse of other periods when GOM forms were observed or omit this section considering the limitations of using two-week samples and 72-hr back trajectories to explain long range transport (beyond three days) and transformations of Hg (in the time scale of hours to days).

Response: Our evidence seems pretty clear based on previous data. We agree it is limited

2. Editorial comments and suggestions C3 Printer-friendly version Discussion paper 1) "Tekran®, and nylon, and cation exchange membranes", please clarify. Response: This has been fixed.

2) L187-192, this paragraph is less relevant to section 3.1, could be omitted or placed in another section. Response: We feel this part is important here.

3) "Overall, air concentrations measured by the Tekran® system in this study were similar to those measured in 2010 (Peterson et al., 2012)." and L250, same site? Response: This has been adjusted.

4) L200-204, please rephrase. Response: This was done.

5) L205, please provide the two forms. Response: This was done.

6) L236-238, please provide the forms at the rural site and traffic site. Response: This was done and we thank the reviewer for this comment for this strengthens our argument for viable used of thermal desorption to identify chemical forms.

7) "Modeled GOM dry deposition fluxes were calculated", "measured Hg dry deposition fluxes were similar to those modeled simulations with modeled GOM dry deposition _=_=2", please rephrase. Response: This has been corrected.

8) L282-283, please rephrase. Response: "However" has been removed.

9) "the profile was a gradual increase", please rephrase. Response: This has been reworded.

10) "In order to improve our understanding of Hg air-surface exchange, and measure GOM physiochemical properties of different GOM compounds need to be understood", please rephrase. Response: This sentence has been removed from the abstract and the conclusions. It is now replaced by the following "In order to measure GOM accurately, we need to know what compounds exist in the atmosphere."

Gertler A, Bennett J. The Nevada Rural Ozone Initiative: A framework for developing an understanding of factors contributing to elevated ozone concentrations in rural and remote environments. Science of the Total Environment 2015; 530: 453-454. Lefohn AS, Cooper OR. Introduction to the special issue on observations and source attribution of ozone in rural regions of the western United States Preface. Atmospheric Environment 2015; 109: 279-281.

Please also note the supplement to this comment:
http://www.atmos-chem-phys-discuss.net/acp-2016-725/acp-2016-725-AC3-supplement.pdf

———————————————————

[Figure]

**Supplement:**

[revised manuscript text omitted]